# Integrated Metabolome and Transcriptome Analysis Provide Insights into the Effects of Grafting on Fruit Flavor of Cucumber with Different Rootstocks

**DOI:** 10.3390/ijms20143592

**Published:** 2019-07-23

**Authors:** Li Miao, Qinghua Di, Tianshu Sun, Yansu Li, Ying Duan, Jun Wang, Yan Yan, Chaoxing He, Changlin Wang, Xianchang Yu

**Affiliations:** 1Institute of Vegetables and Flowers, Chinese Academy of Agriculture Sciences, Beijing 100081, China; 2Beijing Key Laboratory of Growth and Developmental Regulation for Protected Vegetable Crops, College of Horticulture, China Agricultural University, Beijing 100193, China

**Keywords:** grafted cucumber, rootstock, transcriptomic, metabolomic, sugars, flavor

## Abstract

Rootstocks frequently exert detrimental effects on the fruit quality of grafted cucumber (*Cucumis sativus* L.) plants. To understand and ultimately correct this deficiency, a transcriptomic and metabolomic comparative analysis was performed among cucumber fruits from non-grafted plants (NG), and fruits from plants grafted onto different rootstocks of No.96 and No.45 (*Cucurbita moschata.* Duch), known to confer a different aroma and taste. We found remarkable changes in the primary metabolites of sugars, organic acids, amino acids, and alcohols in the fruit of the grafted cucumber plants with different rootstocks, compared to the non-grafted ones, especially No.45. We identified 140, 131, and 244 differentially expressed genes (DEGs) in the comparisons of GNo.96 vs. NG, GNo.45 vs. NG, and GNo.45 vs. GNo.96. The identified DEGs have functions involved in many metabolic processes, such as starch and sucrose metabolism; the biosynthesis of diterpenoid, carotenoid, and zeatin compounds; and plant hormone signal transduction. Members of the HSF, AP2/ERF-ERF, HB-HD-ZIP, and MYB transcription factor families were triggered in the grafted cucumbers, especially in the cucumber grafted on No.96. Based on a correlation analysis of the relationships between the metabolites and genes, we screened 10 candidate genes likely to be involved in sugar metabolism (Fructose-6-phosphate and trehalose), linoleic acid, and amino-acid (isoleucine, proline, and valine) biosynthesis in grafted cucumbers, and then confirmed the gene expression patterns of these genes by qRT-PCR. The levels of *TPS15* (Csa3G040850) were remarkably increased in cucumber fruit with No.96 rootstock compared with No.45, suggesting changes in the volatile chemical production. Together, the results of this study improve our understanding of flavor changes in grafted cucumbers, and identify the candidate genes involved in this process.

## 1. Introduction

Cucumber (*Cucumis sativus* L.) is an economically important horticulture crop, and under unfavorable soil and environmental conditions, its growth and production are commonly promoted by grafting [1]. However, the specific rootstock used can considerably affect the levels of flavor compounds and the overall fruit quality [2]. Yield and quality are often considered competing traits [3]. To improve the productivity and economic benefits, breeders and producers typically ignore or sacrifice crop quality during rootstock selection and breeding [4]. Quality is a particularly problematic term because of the lack of objective criteria for evaluation. Additionally, there is an absence of fundamental knowledge concerning chemical-driven consumer preferences, the pathways for their synthesis, and the genes regulating the output of these pathways [2]. Hence, the overall molecular mechanism of fruit quality affected by grafting remains largely unknown. With an increased consumer demand for quality, understanding the implications of grafting for fruit quality, and elucidating the involved mechanisms, becomes imperative.

Fruit quality is affected by grafting in many aspects, including the morphometric and textural characteristics, sweetness and acidity, and aroma profile (Figure 1) [4]. The latter two parts could be considered together as flavor, a feature that has been attracting increasingly stronger research and consumer attention. Historically, the evaluation of vegetable flavor started with the identification of its constituents, including the contents of acids, sugars, volatiles, and other compounds, which individually elicit sensory responses [2]. The soluble solids content (SSC) and fructose were found to be significantly reduced in cucumber fruits collected from many growers in cucumber growing areas from plants using different rootstocks [5]. The SSC and titratable acidity (TA) were higher in both the grafted and self-grafted cucumber in the presence of 60 mM of NaCl [6]. The appropriated value of the TSS/TA ratio represents a central parameter, as it describes a good balance between sweetness and acidity in fruits [2]. However, the aroma and taste of the fruits depend not only on the scion, but also on the rootstock, and changes in the fruit quality after grafting depend on the different rootstocks used [7]. However, the molecular mechanisms by which grafting alters sweetness and acidity have remained largely unknown.

Volatile organic compounds (VOCs) play a critical role in fruit quality. A total of 85 volatile chemicals were identified in 23 different tissues of cucumber plants using solid-phase microextraction combined with gas chromatography–mass spectrometry, including 36 volatile terpenes [8]. Aldehydes and alcohols were the main sources of the aromatic flavor of fruits of different cucumber varieties [9]. More specifically, the unsaturated aldehydes (E, Z)-2,6-nonadienal and (E)-2-nonenal, are the most important and abundant VOCs in cucumbers [10], but (Z)-6-nonenol and (E, Z)-2,6-nonadienal were the most abundant compounds in cucumber peel and flesh, respectively, and the percentage composition of VOCs in the peel and flesh of cucumber was significantly affected by grafting [11]. Two (3Z):(2E)-hexenal isomerases (HIs) from cucumber fruits are responsible for the rearrangement of Z-3-aldehydes in both the leaves and fruits, and also play important roles in the formation of VOCs in cucumber fruits [12]. Although the enzymes responsible for the metabolism of plant VOC production have been extensively studied [8], less is known about the molecular mechanism of the divergence of VOC production in different rootstocks.

Recently, transcriptomics and metabolomics have been applied as powerful tools to explain the flavor traits in many plants, such as tomato [13], cucumber [8], tea [14], and grape [15]. Through whole-genome sequencing and measuring the flavor associated chemicals, the genetic loci that affect most of the target flavor chemicals in tomatoes, including sugars, acids, and volatiles, were identified in 398 modern, heirloom, and wild accessions [13]. The transcriptomic and metabolomic analyses revealed that the genes related to theanine and caffeine metabolism were significantly downregulated, and the genes related to the flavonoid biosynthetic pathway were mostly upregulated in grafts of *Camellia sinensis* and *C. oleifera* [14]. Recently, Zhao et al. (2018) [7] found that different rootstocks, figleaf gourd rootstock (*Cucurbita ficifolia*), and “Weisheng No.1” (*Cucurbita moschata* × *Cucurbita moschata* hybrids) affected the scion flavors in the physiological profiling and the transcripts of sugar and aromatic flavor-related genes in the cucumber mesocarp. These transcriptomic analyses increased the understanding of flavor production at the metabolic level. However, the precise mechanism of the effect of grafting on the fruit flavor of cucumber is still not clear, especially the mechanisms driven by different rootstocks from the same species.

Compared with *Cucurbita ficifolia bouche*, *C.moschata* has better characteristics and has been exploited as a favorable source of rootstock [16,17]. We selected No.45 and No.96 from 102 *Cucurbita moschata* varieties of rootstock, based on the grafted cucumber growth characteristics, physiology traits, fruit quality, and the significantly different fruit flavor of the cucumber grafted onto the two rootstocks [18]. According to these results, we applied integrated metabolomics and transcriptomics tools to explore the candidate genes involved in grafted cucumber flavor production through metabolomic-gene correlation analysis. Specifically, cucumbers were grafted onto two rootstocks, No.45 and No.96 (*C. moschata*), whereas non-grafted plants were used as the control. To determine the difference in fruit quality in the cucumber plants grafted on the two rootstocks, we first measured the quality characteristics of the cucumber fruit, including the brix, acids, wax power, and nitrate content. Next, the flesh of the cucumber grafted on different rootstocks was subjected to gas chromatography tandem time-of-flight mass spectrometry (GC-TOFMS) and RNA sequencing (RNA-seq). Finally, the metabolites and genes correlated with flavor were identified, and the important components involved in sugar metabolism, linoleic acid metabolism, and amino-acid biosynthesis, which were significantly affected by the different rootstocks, were screened.

## 2. Results

### 2.1. Phenotype and Physiological Traits of Cucumber Fruit from Plants Grafted on Different Rootstocks

At commercial maturity, significant morphological differences were observed in the fruit of the cucumber grafted on different rootstocks. Compared with non-grafted cucumber, there were lower levels of wax power on the surface of the cucumber fruit grafted onto *Cucurbita moschata* rootstocks, and the cucumber fruit grafted onto the No.96 rootstock lacked a wax power layer and appeared glossy (Figure 2A); this was consistent with the measurements of the wax power content on the surface of the fruit by colorimeter (Figure 2C). Compared with the non-grafted cucumber, grafting significantly decreased the contents of the free amino acid (FAA), soluble protein, and nitrate, and increased the soluble solid contents and acidity in the cucumber fruits grafted onto the No.45 rootstock. Compared with the non-grafted plants, a lower soluble protein content and nitrite content, and higher total acid content were observed in the cucumber fruits grafted onto the No.96 rootstock, but there were not significant effects on the soluble solid and FAA contents in the cucumber fruits of these plants. Compared with the cucumber grafted onto No.96, the cucumber grafted onto No.45 showed increased amounts of soluble solid and nitrate, and decreased amounts of FAA and soluble protein. The titratable acid of the cucumber fruit was not significantly different between the No.45 and No.96 rootstocks (Figure 2C). These changes in the physiological characteristics indicated that grafting significantly affected the quality of the cucumber fruit, and the cucumber flavor was remarkably different between the rootstocks of *Cucurbita moschata* No. 45 and No. 96.

### 2.2. The Metabolite Profiles Revealed Differences in Metabolic Regulation between Non-Grafted and Grafted Cucumber Plants

A total number of 373 peaks were identified, and 169 metabolites were annotated (Appendix A), each with four biological replicates. Unsupervised principal component analysis (PCA; Appendix A) was performed before supervised OPLS-DA. The results showed that the metabolites measured in the fruit of GNo.45 and NG plants, GNo.96 and NG plants, and GNo.45 and GNo.96 plants were, respectively, clustered together in the OPLS score plots (Figure 3). Setting the variable importance in the projection (VIP) as ≥1.0, together with a *p*-value of ≤0.5, was used as the threshold for significant differences. The contents of 34, 8, and 21 metabolites were significantly different in the comparisons of GNo.45 vs. NG, GNo.96 vs. NG, and GNo.45 vs. GNo.96, respectively. The different metabolites (DMs) are shown in Figure 3C, and include sugars and derivatives, amino acids, organic acids, and alcohols. Most of the DMs in organic acids and amino acids are involved in the TCA cycle or amino-acid biosynthesis, such as citric, succinic acid, isoleucine, and phenylalanine (Figure 3C). Compared with non-grafted cucumber, the amounts of fructose, tartronic acid, and 2-hydroxy-3-isopropylbutanedioic acid increased 7.3, 8.28, and 19 times, respectively, in the cucumber fruit grafted onto No.45. Additionally, the amounts of 2,4-diaminobutyric acid, phenylalanine, 2-hydroxy-3-isopropylbutanedioic acid, and digitoxose increased 3.8, 4.2, 14, 422, and 464 times, respectively, in the cucumber fruit grafted onto No.45, compared with the cucumber grafted onto No.96.

### 2.3. Identified Differentially Expressed Genes (DEGs) and Their Gene Ontology (GO) Enrichments by Transcriptome Analysis

We performed RNA-seq profiling with the same plant materials that were used for the GC-MS profiling. Totals of 131 (with 88 upregulated and 43 downregulated), 140 (with 111 upregulated and 29 downregulated), and 244 (with 126 upregulated and 118 downregulated) differentially expressed genes (DEGs) were identified in comparisons of GNo.45 vs. NG, GNo.96 vs. NG, and GNo.45 vs. GNo.96, respectively. Among these DEGs, only 17 genes were identical in both XT/No.45 and XT/No.96, compared with those in the non-grafted cucumber, and 48, 66, and 116 were unigenes in the comparisons of GNo.45 vs. NG, GNo.96 vs. NG, and GNo.45 vs. GNo.96, respectively (Figure 4A,B). These results indicated differences at the molecular level between the grafted and non-grafted cucumbers, and also between cucumber fruits grafted onto different rootstocks.

We employed GO assignments to classify the functions of the DEGs of the cucumber fruit grafted onto different rootstocks compared with non-grafted cucumber, and cucumber fruit grafted onto No.45 compared with No.96 (Figure 4C). The DEGs from the comparisons of GNo.45 vs. NG, GNo.96 vs. NG, and GNo.45 vs. GNo.96, were significantly enriched in the terms of “cell”, “cell part”, and “organelle”. In the molecular function category, the DEGs reveal significant enrichment in categories for “binding” and “catalytic activity”. In addition, comparison of NG vs. GNo.96 also revealed significant enrichment for “nucleic acid binding transcription factor activity”, and “molecular transducer activity” terms. In the biological process category, the “metabolic process”, “single-organism process”, “cellular process”, “response to stimulus”, and “biological regulation” were significantly enriched. A GO analysis of the DEGs indicated that they were functionally enriched in diverse biological processes. Some DEGs are involved in the key metabolism, which remarkably affects fruit quality in the grafted cucumber, including carbohydrate metabolism, glycolytic process, glyoxylate cycle, and fatty acid catabolic process.

### 2.4. Clustering of Expression Patterns and Transcription Factors in Analysis of Transcriptome Data

The significant DEGs were clustered into four clusters by K-means clustering, based on the Pearson correlation distances (Figure 5). Each cluster represented the gene expression of each sample. Cluster 1 contained the genes that were particularly downregulated in the heterograft GNo.45 in comparison to the non-grafted control and heterograft GNo.96. The functional categories were associated with starch and sucrose metabolism (Csa1G611290; Csa3G402970; Csa5G568300; Csa7G343850), diterpenoid biosynthesis (Csa3G903550; Csa3G904060), cyanoamino acid metabolism (Csa1G611290; Csa3G402970), phenylpropanoid biosynthesis (Csa1G611290; Csa3G402970; Csa4G038620), and pentose and glucuronate interconversions (Csa5G568300; Csa7G343850). Cluster 2 contained genes that were particularly upregulated in the heterograft GNo.45 in comparison to the self-grafted control and heterograft GNo.96. The functional category of the photosynthesis-antenna proteins, carotenoid biosynthesis, and diterpenoid biosynthesis were enriched in this cluster, along with the GO term catalytic activity. The genes related to zeatin biosynthesis and carotenoid biosynthesis exhibited a higher expression in the heterograft GNo.96 cucumber fruit than in the heterograft GNo.45 and non-grafted cucumber (Cluster 3). The genes involved in plant hormone signal transduction and RNA degradation were expressed at significantly lower levels in the heterograft GNo.96 cucumber fruit than the heterograft GNo.45 and non-grafted cucumber (Cluster 4).

Transcription factors (TFs) regulate plant-specific genes and/or mediate a variety of plant-specific signals in many biological and biochemical processes [19,20]. We identified the cucumber TFs by comparing the approximate sequences of TFs from the plant TF database (http://planttfdb.cbi.pku.edu.cn). A total number of 2519 candidate unigenes with a high-sequence similarity to the TFs were identified (Appendix A). The expression analysis of these candidate TFs revealed that 17 and 31 TFs were differentially expressed in GNo.45 and GNo.96, respectively, compared with non-grafted cucumber, and the most represented DEGs were members of the HSF and AP2/ERF-ERF TF family, respectively. Thirty-five TFs were differentially expressed in GNo.96 compared with GNo.45, and the most abundant TFs of the DGs were members of the AP2/ERF-ERF TF family (Figure 6). Most of these TF families showed notable upregulation in the grafted cucumber fruits compared with the autografted plants (Appendix A).

### 2.5. Candidate Genes for Metabolites of Sugar Metabolism, Linoleic Acid Metabolism, and Amino-Acid Biosynthesis

To clearly determine the association between the DEGs and DMs, we integrated the transcriptome and the metabolite data by correlation analysis. Ten DEGs that are likely involved in sugar metabolism, linoleic acid metabolism, and amino-acid biosynthesis, were identified based on previous knowledge and the setting of 0.05 > Pearson correlation coefficient (PCC) > 0.8 (Table 1). Under this evaluation condition, we observed a significantly higher fructose-6-phosphate content in GNo.45 than in the non-grafted cucumber, which was consistent with the expression levels of fructose-bisphosphate aldolase (Csa2G252020). The starch and sucrose metabolism genes were also upregulated in GNo.45 compared with those in GNo.96, which was in agreement with the soluble solid contents. Linoleic acid is one of the main products of fatty acid metabolism, and is also the substrate of fatty acid oxidation that is involved in the synthesis of volatile compounds [21,22]. In our study, we found that the linoleic acid content significantly increased in GNo.45, but decreased in GNo.96 in comparison with the non-grafted cucumber fruits. This was also consistent with the changes of the correlated genes lipoxygenase (Csa2G023880) in the grafted cucumber under a different rootstock. However, the candidate genes likely involved in VOCs production exhibited no changes in the grafted cucumber fruit compared with the non-graft cucumber, except for TPS15 (Csa3G040850; Appendix A) [8]. Additionally, we found that amino-acid biosynthesis genes, which play positive roles, were upregulated, and the genes that play negative roles were downregulated in GNo.45 compared with non-grafted cucumber; as expected, these DEGs also correlated with the levels of isoleucine, proline, and valine (Table 1, Figure 7). Consequently, these DEGs are valuable potential targets for changing the flavor quality of grafted cucumber fruit.

### 2.6. Validation of Transcriptomic Data Accuracy by qRT-PCR

To validate the RNA-seq results, 16 genes were selected randomly, and the RNA levels were tested by qPCR. The fragments per kilobase of transcript per million (FPKM) mapped reads values were well correlated with the relative expression levels determined via qPCR for three biological replicates (Figure 8). This consistency of the RNA-seq data and RT-qPCR indicated that the data could be used to assess the up- and down-regulated gene expression.

## 3. Discussion

Grafting is used in cucumber cultivation, because the well-developed root of the rootstock is often resistant or tolerant to biotic or abiotic stress [4,16]. Nonetheless, the use of grafting can result in deterioration in the flavor quality of grafted cucumbers, decreasing consumer value. In this study, we used a combined metabolome and transcriptome approach to effectively explore the candidate metabolites, pathways, and genes affecting grafted cucumber flavors (Figure 9).

### 3.1. Fruit Characteristics of Non-Grafted Cucumber and Cucumbers Grafted onto Different Rootstocks

Genus Cucurbita is the most popular rootstocks for cucumbers. *Cucurbita ficifolia* Bouche exerted detrimental effects on the cucumber fruit quality, and *Cucurbita moschata* Duch. has been recently exploited as a favorable source of rootstocks [16,17]. Selecting appropriate rootstocks is critical to determining the changes in the grafted cucumber aroma and taste. We selected No.45 and No.96 as experimental materials from 102 *Cucubita moschata* varieties based on the SSC, wax power content, and previous evaluations [18]. We found that the levels of wax power on the surface of the cucumber fruit were significantly decreased in GNo.96 compared with GNo.45 (Figure 2A). Many previous studies revealed that the SSC, fructose concentration, and titratable acidity differed for different rootstocks under various conditions [5,6]. Nitrates in vegetables may be beneficial for human health [23]. Nitrate/nitrite is a vital flavor quality in cucumber, especially in the peel [24]. In our study, the wax power content, SSC, FAA content, soluble protein, and nitrate content in the cucumber fruit were significantly different, and the titratable acidity was similar between GNo.45 and GNo.96. The use of two typical rootstocks with notable effects on the grafted cucumber fruit quality allowed for thw study of the molecular basis of flavor changes related to grafting.

### 3.2. Different Transcriptomic and Metabolomic Regulation in Grafted Cucumber Fruit for Different Rootstocks

To investigate the fruit quality, we tested cucumber samples without the peel. The metabolomic data showed that the cucumber fruit quality was notably affected in No.45 compared with No.96. Most DMs are sugars, organics acids, amino acids, and alcohols (Figure 3D). The total concentration of soluble carbohydrates and the relative proportions of the three main sugars of glucose, fructose, and sucrose, contribute differentially to sweetness, with fructose considered to be the sweetest [7,25,26]. In our study, the contents of sucrose, fructose, and fructose-6-phosphate significantly increased in the fruit of GNo.45 compared with the non-grafted ones, with a 7.3 increase in the fructose contents. This was consistent with the SSC content in the fruit of GNo.45. This suggested that the sucrose and fructose determined the sweetness of the cucumber fruits grafted onto No.45, but not No.96. Previous work indicated that a variation in fruit sweetness depended on different combinations of scion and rootstock [6,27]. In addition, digitoxose in the cucumber of GNo.45 was increased 422,646 times that of the level found in GNo.96. D-glucose is the most effective precursor of digitoxose in most glycosides, including D-glucose, D-fructose, D-mannose, and D-galactose [28]. However, the digitoxose function in cucumber fruit is not clear. The transcriptome analysis showed that *Cucurbita moschata* No.45 and No.96 similarly affected the cucumber fruit quality compared to non-grafted cucumber, and the effect of these two rootstocks on the cucumber fruit was notably different at the molecular level. The enriched GO terms based on the DEGs confirmed the divergent molecular processes involved in the cucumber fruit quality changes by the two rootstock cultivars, such as “metabolic process”, “single-organism process”, “cellular process”, “response to stimulus”, and “biological regulation”. Sugar metabolism, diterpenoid biosynthesis, plant hormone signal transduction, and carotenoid biosynthesis were the significant metabolism processes that showed differences by KEGG and common expression pattern analysis (Appendix A). These results indicated that the changes in cucumber fruit flavor as a result of different rootstocks occur in many biological processes. However, the DEGs identified in this study were not in accordance with reports by Zhao et al. [7], possibly because of the use of two rootstocks from the same species in this study.

### 3.3. Volatile Organic Compounds and Transcript Factors Analysis

Volatile organic compounds (VOCs) are an important element of a plant’s chemotype, acting to protect plants from diverse unfavorable environment conditions and improve pollination, as well as playing a critical role in the formation of fruit flavor [8,22]. Based on the biosynthetic origins, VOCs can be roughly divided into three main classes, namely: terpenoids, which originate from mevalonic acid (MVA) and methylerythritol phosphate (MEP); phenylpropanoid/benzenoid compounds, which originate from the aromatic amino acid phenylalanine (Phe); and aldehydes/acids/alcohols/esters, which originate from C_18_ unsaturated fatty acids, linoleic or linolenic acid, and branched-chain amino acids [22]. Currently, unsaturated aldehydes (E, Z)-2,6-nonadienal and (E)-2-nonenal are considered the most important VOCs in cucumber, because of their high abundances [29,30]. Cucumber grafted onto bottle gourd can exhibit changes in the percentage composition of the major VOCs in the peel and flesh [11]. Linoleic acid is the precursor for a variety of fatty acid-derived VOCs via the 13- and 9-lipoxygenase (13-LOX and 9-LOX) pathways [21]. In our study, linoleic acid significantly increased in cucumber fruit grafted onto No.45, and decreased in cucumber grafted onto No.96. Many amino acids differed in the cucumber fruit grafted on to different rootstocks, including isoleucine (Figure 3D). Additionally, the DEGs in GNo.45 compared with non-grafted plants, were enriched in “diterpenoid biosynthesis”, “phenylpropanoid biosynthesis”, and “carotenoid biosynthesis”, all involved in the generation of VOCs [8]. The VOC compositions may have changed in the cucumber grafted onto *Cucurbita moschata* No.45 and No.96 rootstocks. The candidate genes previously proposed to be involved in the cucumber VOC production, based on detection in 23 different tissues of the cucumber plants through integrative analyses of non-targeted volatile profiling and transcriptome data [8], were not notably changed in the grafted cucumber fruit in our study, except for TPS15 (Csa3G040850). This possibly reflects variety-specific and grafting effects. Therefore, the changes of VOCs in the fruits of grafted cucumber plants required further investigation.

Transcription factors (TFs) are critical proteins in the regulation of gene expression and signal transduction networks during plant growth and development. The expression levels of the transcription factor genes were significantly influenced after grafting in cucumber [7], tea [14], and grapevine [31]. The three largest families of transcription factors, AP2/EREBP (APETALA2/ethylene responsive element binding protein), MYB-(R1)R2R3, and bHLH, play a variety of roles throughout the plant life-cycle [32]. A recent report indicated that color and flavor are closely linking by job-sharing the same basic-helix-loop-helix (bHLH) TFs called *Noemi* [33]. In our study, the DEGs of comparison of NG vs. GNo.96 reveals a significant enrichment of “nucleic acid binding transcription factor activity” and “molecular transducer activity” terms by GO analysis. The AP2/EREBR protein family includes ethylene-responsive element-binding factors that are involved in plant abiotic stress responses [34], and that control fruit color during tomato ripening [35]. Members of this family were the most abundant TF DEGs in GNo.96 compared with self-grafted cucumber fruit. HSF is the most important transcriptional family for plant response to heat [36], and was notably trigged by grafting in the fruit of GNo.45. Other factors, such as HB-HD-ZIP, NAC, and MYB, also showed significant changes in grafted cucumber fruit. These findings are consistent with the results obtained by Zhao et al. [7], suggesting the involvement of transcription factors in the regulation of cucumber fruit flavors of plants grafted onto different rootstocks.

### 3.4. Metabolite-Based Determination of Candidate Genes for Fruit in Grafted Cucumber

Many studies have shown that cucumber fruit quality is associated not only with the scion, but also the rootstock. Fruit SSC, fructose concentration, acids, and volatiles vary among different scion-graft combinations [4]. Using transcriptome analysis, sugar and aromatic flavor genes were identified in cucumber fruits from the plants grafted onto figleaf gourd rootstock (*Cucurbita ficifolia*) and Weisheng No.1 (*Cucurbita moschata* × *Cucurbita moschata* hybrids) [7]. Glycolysis/gluconeogenesis metabolism is critically involved in starch and sugar biosynthesis, and the DEGs related to the enzymatic reactions in this pathway are responsible for the starch and sugar content in the cucumber fruit grafted onto different rootstock [7,37]. Fructose metabolism and α-linolenic acid metabolism in the cucumber fruit were also significantly affected by the rootstock at a molecular level [7]. Additionally, TPS15 had been demonstrated to be responsible for volatile terpenoid production in the fruit tissues of cucumber plants by in vitro biochemical experiments [8], as seen in our study. Amino acids alanine, valine, leucine, and methionine are generated from glycolysis and the TCA cycle, and produce many volatile compounds, especially those highly abundant in floral scents and fruit aromas [22].

We investigated the changes in the cucumber fruits from the plants grafted on two rootstocks, with significant differences in physiological profiling, as reflected by metabolic and transcriptome analysis. We focused our attention on the primary carbon metabolism, including glycolysis and the TCA cycle, based on the metabolic-gene correlation analysis results (Figure 7). Glycolysis and the TCA cycle not only provide an energy and carbon source for plant growth and development, their derivatives also affect the quality of fruit, such as sweetness and aroma profile [22]. In total, seven DMs related to the glycolysis metabolism were notably increased in GNo.45 compared with the non-grafted cucumber, including sucrose and fructose. Fructose is one of the major free sugars in cucumber fruit [38], and is regarded as the sweetest sugar. The expression of fructose-bisphosphate aldolase (Csa2G252020) was also significantly higher in GNo.45 than in the self-grafted plants. Six related DEGs exhibit different expression patterns in cucumber grafted onto different rootstocks, including two beta-glucosidases (Csa3G402970 and Csa1G611290), two pectin esterases (Csa5G568300 and Csa7G343850), and two trehalose-6-phosphate phosphatases (Csa4G028470 and Csa6G500540). In plants, the glycoside hydrolase (GH) family 1 beta -glycosidases are thought to be critically involved in many diverse processes, including chemical defense against herbivory, lignification, hydrolysis of cell wall-derived oligosaccharides during germination, and control of the active phytohormone levels [39]. Pectin esterases (PE) catalyze the demethylation of pectin, and the total PE activity decreased at the ending of the fruit ripening process in many plants, such as melon [40], peach [41], and strawberry [42]. Fruit-specific PE isoform affects tissue integrity during senescence in tomato [43]. Trehalose-6-phosphate phosphatase (TPP) is a critical enzyme for the synthesis of Trehalose-6-phosphate phosphatase, which regulates sugar influx into glycolysis [44]. Putative shikimate kinase (Csa6G487590) and LL-diaminopimelate aminotransferase (Csa2G404730) are involved in the biosynthesis of amino acids. Additionally, lipoxygenase (Csa2G023880) is possibly responsible for linoleic acid metabolism. The lipoxygenase (LOX) pathway induces unsaturated fatty acids to undergo stereospecific oxygenation to form 9-hydroperoxy and 13-hydroperoxy intermediates, finally yielding volatile compounds [45]. The expression patterns of these candidate genes were verified with quantitative RT-PCR (Appendix A), except for Csa7G343850 and Csa6G487590. These candidate genes should be helpful to improve grafted cucumber fruit quality, but functional studies in candidate genes are still required in order to clarify the differences induced by the rootstock.

## 4. Materials and Methods

### 4.1. Plant Materials and Growth Conditions

This study was carried out on the farm of the Institute of Vegetables and Flowers of the Chinese Academy of Agricultural Sciences, Beijing, China. Cucumber cultivar “xin tai mi ci” (*Cucumis sativus* L.) was grafted onto two pumpkin cultivars (*Cucurbita moschata*), “GNo.45” and “GNo.96”. The seeds of the scions and rootstocks were sown in 50-cell and 32-cell polystyrene trays (54 × 28 × 5 cm), respectively, containing commercial organic substrates (Vpeatmoss: Vvermiculite: Vperlite = 1:1:1). The germination environmental conditions were 25–28 °C and 85–90% relative humidity. Hole insertion grafting was conducted, and the grafted seedlings were maintained using conditions described in previously published protocols [18]. Non-grafted cucumber seedlings were used as the control (NG). Cucumber fruits without peel were harvested 10 days after anthesis (DAA), the point of commercial maturity, from the 12th to the 15th node, after carefully recording the numbers of the nodes (Figure 2B). One part of the sample was used for physiological character determination, the other parts were quickly frozen using liquid nitrogen, and then fruit without peel was cut into small pieces for separate GC-MS analysis and RNA-seq. Tissue samples were collected from at least three individual plants, and six and three biological replicates were used in the GC-MS analysis and RNA-seq, respectively.

### 4.2. Determination of Contents of Soluble Solid, Wax Power, FAA, Soluble Protein, Nitrate, and Acidity

The middle parts of the cucumber fruits without peel were transversely cut into slices, and four to five slices were squeezed to obtain juice, which was analyzed using a digital refractometer (model PR-100, ATAGO, Tokyo, Japan). A total of five to seven fruits were harvested based on the availability of the plant materials. Then, the fruits were cut into 12 equal pieces. Three pieces were randomly selected and separately analyzed to determine their soluble solid contents (SSC) using the Brix assay, and the remaining pieces were subjected to sensory evaluation [18].

In the fruiting stage, five commercial ripe fruits that flowered at the same time were selected from each treatment, and the brightness of the fruit surface was measured with a colorimeter as an indicator of the level of wax power occurrence. We randomly selected three points in the fruit, at the front, middle, and tail, and determined the brightness value of L1 according to the lightness of a tape. The tape without wax power measured a brightness value of L0. The lightness difference (L = L1–L0) value was used to indicate the level of wax power occurrence. Each site was measured three times, and the average value was calculated.

The middle sections of the cucumber fruit samples (0.5–2 g fresh weight) were harvested to measure the content of soluble proteins, total free amino acids, nitrate content, and titratable acidity. The soluble proteins were determined by the G-250 reagent method, as described by Bradford [46]. The total free amino acids were measured using the ninhydrin method [47]. For the nitrate measurement, we employed the modified visible spectrophotometry method [48]. Titratable acidity was measured by the methods described by Moradshahi [49]. Three to five fruits from each treatment were measured, with three biological replicates.

### 4.3. Sample Extraction and GC-MS Analysis

The cucumber samples were sealed in a 50 mL tube and were stored at −80 °C for no longer than 1 week before extraction and analysis.

For analysis, 60 ± 1 mg samples were measured into 2-mL EP tubes and were extracted with 0.48 mL extraction liquid (V_Methanol_: V_H2O_ = 3:1). Next, 10 μL of adonitol (0.5 mg/mL stock in dH_2_O) was added as an internal standard, and was vortex mixed for 30 s. The samples were then homogenized in a ball mill for 4 min at 45 Hz, then treated with ultrasound for 5 min (incubated in ice water) and centrifuged for 15 min at 12,000 rpm at 4 °C. Next, the supernatant (0.35mL) was transferred into a fresh 2-mL GC/MS glass vial, and 60 μL from each sample was pooled as a QC sample. After drying completely in a vacuum concentrator without heating, 120 μL methoxy amination hydrochloride (20 mg/mL in pyridine) was added, and then the sample was incubated for 30 min at 80 °C. Next, 160 μL of the BSTFA regent (1% TMCS, *v*/*v*) was added, and the sample aliquots were incubated for 1.5 h at 70 °C. Next, 10 μL FAMEs (standard mixture of fatty acid methyl esters, C8–C16: 1 mg/mL; C1–C24: 0.5 mg/mL in chloroform) was added to the QC sample when cooled to room temperature.

All of the samples were then analyzed using an Agilent 7890 gas chromatograph system coupled with a Pegasus HT time-of-flight mass spectrometer (GC-TOFMS). The system utilized a DB-5MS capillary column coated with 5% diphenyl cross-linked with 95% dimethylpolysiloxane (30m × 250 μm inner diameter, 0.25 μm film thickness; J&W Scientific, Folsom, CA, USA). A 1μL aliquot of the analyte was injected in splitless mode. Helium was used as the carrier gas, the front inlet purge flow was 3 mL min^−1^, and the gas flow rate through the column was 1 mL min^−1^. The initial temperature was kept at 50 °C for 1 min, then increased to 310 °C at a rate of 10 °C min^−1^, and was maintained for 5 min at 310 °C. The injection, transfer line, and ion source temperatures were 280, 280, and 250 °C, respectively. The energy was −70 eV in the electron impact mode. The mass spectrometry data were acquired in full-scan mode, with the m/z range of 50–500 at a rate of 20 spectra per second after a solvent delay of 6.27 min. Four independent biological replicates were analyzed.

### 4.4. RNA Extraction, cDNA Library Preparation, and RNA Sequencing

The total RNA samples were isolated, as previously described (Miao et al., 2019). A total amount of 3 μg of the RNA per sample was used for the cDNA library preparation after the extracted RNA passed a quality evaluation. Sequencing libraries were constructed by NEBNext^®^ UltraTM RNA Library Prep Kit for Illumina^®^ (NEB, Ipswich, MA, USA) following the manufacturer’s instructions. Then, the prepared library was sequenced on an Illumina HiSeq 2000 platform (San Diego, CA, USA).

### 4.5. Analysis of Differentially Expressed Genes (DEGs) and Functional Annotations

We aligned the cleaned reads to the reference index by HISAT2. Unique mapped reads were filtered with a mapping quality threshold over 15 (*p* < 0.05). String tie was used to assemble the transcripts according to the cucumber annotation. The gene expression level was calculated based on fragments per kilobase of transcript per million (FPKM) mapped reads. We used FPKM >1, Fc >2, and FDA <0.05 as the thresholds for the significance of the differential gene expression (DEGs) of the scion/rootstock comparison at No.45 vs. NG, No.96 vs. NG, and No.45 vs. No.96. We conducted enrichment analyses of the gene ontology (GO) and KEGG pathways based on all of the DEGs. The transcriptions factors were predicted by HMMER software.

### 4.6. Validation of Gene Expression by Quantitative Real-Time RCR (qRT-PCR)

The RNA preparations were extracted in triplicate with three biological replications, as described above. Twenty genes were randomly selected for the validation of the RNA-seq results using qRT-PCR. All of the gene-specific primers were designed by the Primer 5.0 program (Appendix A), and th qRT-PCR reactions were performed using a DyNAmo Flash SYBR Green qPCR kit (Thermo Fisher Scientific, Inc., Waltham, MA, USA) and the CFX96 qPCR System (Bio-Rad Hercules, CA, USA). The expression levels were calculated by the 2^−ΔΔ*C*t^ method, and were normalized to that of the reference gene Csa*TIP41* [50].

### 4.7. Statistical Analysis

The expression and metabolite data were standardized as Z-scores through Log2. These data were used for PCA and the calculation of PCCs. We used PCC >0.08 and PCCP <0.05 as the threshold of significance of correlation genes-metabolism for the comparisons of GNo.45 vs. NG, GNo.96 vs. NG, and GNo.45 vs. GNo.96. The physiological data were presented as mean ± standard deviation of the three replicate samples. The data were analyzed with variance (ANOVA) using SAS software. The differences between the treatments were determined by the least significant differences test at *p* < 0.05. The presented figures were generated by Microsoft Excel 2018.

## 5. Conclusions

In the present study, we analyzed the fruit of the non-grafted cucumber, grafted onto different rootstocks (*Cucurbita moschata*) at the morphological, physiological, and metabolic and transcriptome level in order to identify the genes and pathways that contribute to cucumber fruit flavor. We found that the content of the wax power layer of the cucumber fruit was decreased by grafting, especially in GNo.96. The sugar, organic acids, amino acids, and alcohols in cucumber fruit were significantly increased after grafting onto No.45, but showed little difference between GNo.96 and NG. Many DEGs of the data comparisons of non-grafted and grafted cucumber were significantly enriched in the sugar metabolism, carotenoid biosynthesis, and plant hormone signal transduction. Ten DEGs involved in sugar metabolism (fructose-6-phosphate and trehalose), linoleic acid, and amino-acid (isoleucine, proline, and valine) biosynthesis were identified as candidate genes by DMs-gene correlation analysis, which possibly play important roles in grafted cucumber fruit (Figure 9). However, the key metabolites, pathways, and genes for influencing grafted cucumber fruit flavor should be further studied in more rootstock genotypes.

## Figures and Tables

**Figure 1 ijms-20-03592-f001:**
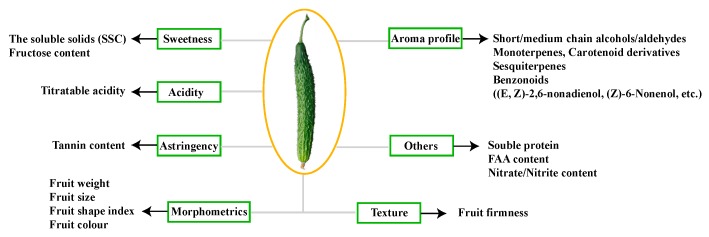
Overview of the major factors determining the quality of cucumber fruit. FAA—free amino acid.

**Figure 2 ijms-20-03592-f002:**
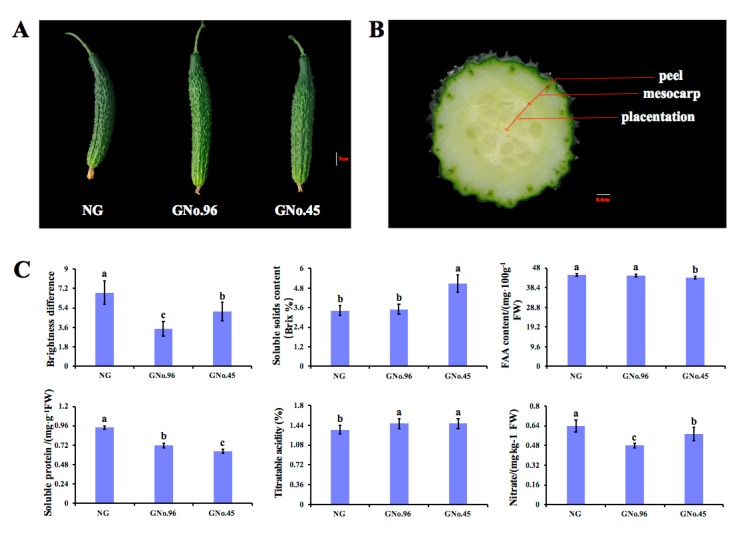
The phenotype and physiological quality of the cucumber fruit in the non-grafted plants, plants grafted onto the rootstocks of No.45 and No.96. (**A**) Phenotype of fruit in non-grafter plants (NG), GNo.96, and GNo.45. (**B**) Peel, mesocarp, and placentation of cucumber fruit. (**C**) Wax power, soluble solids content (SSC), free amino acid (FAA) content, soluble protein content, nitrate content, and titratable acidity in cucumber fruit.

**Figure 3 ijms-20-03592-f003:**
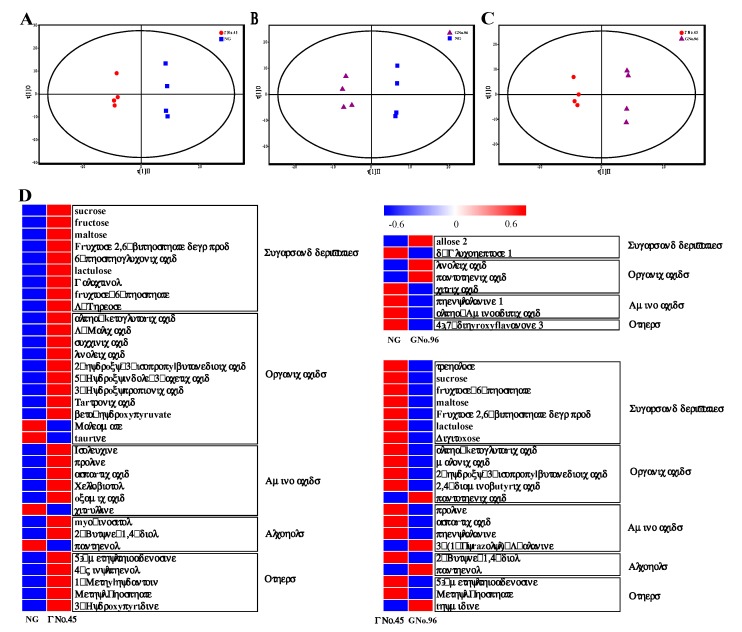
(**A**) OPLS-DA score plots generated from OPLS-DA models and different metabolites (DMs) in the cucumber fruit of NG, GNo.96, and GNo.45. (**B**) OPLS score plots (R2Y = 0.98; Q2 = 0.35). OPLS score plots (R2Y = 0.98; Q2 = 0.12). (**C**) OPLS score plots (R2Y = 0.98; Q2 = 0.27). (**D**) Heat maps of the contents of DMs in sugars and derivatives, organic acids, amino acids, alcohols, and others in fruits of NG, GNo.96, and GNo.45.

**Figure 4 ijms-20-03592-f004:**
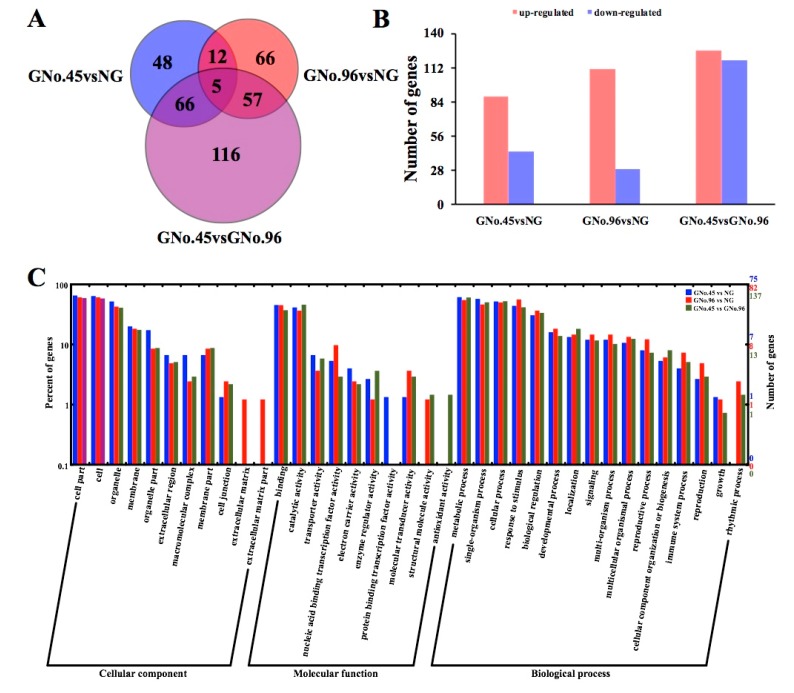
Number and gene ontology (GO) classification enrichment of differentially expressed genes (DEGs) among different comparisons of GNo.45 vs. NG, GNo.96 vs. NG, and GNo.45 vs. GNo.96. (**A**) Venn-diagram of significantly different DEGs. (**B**) Number of up- and down-regulated DEGs. (**C**) GO classification significant enrichment was analyzed by pairwise comparisions of GNo.45 vs. NG, GNo.96 vs. NG, and GNo.45 vs. GNo.96. (*p* < 0.05). The DEGs were assigned into three classifications of the cellular components, molecular function, and biological process.

**Figure 5 ijms-20-03592-f005:**
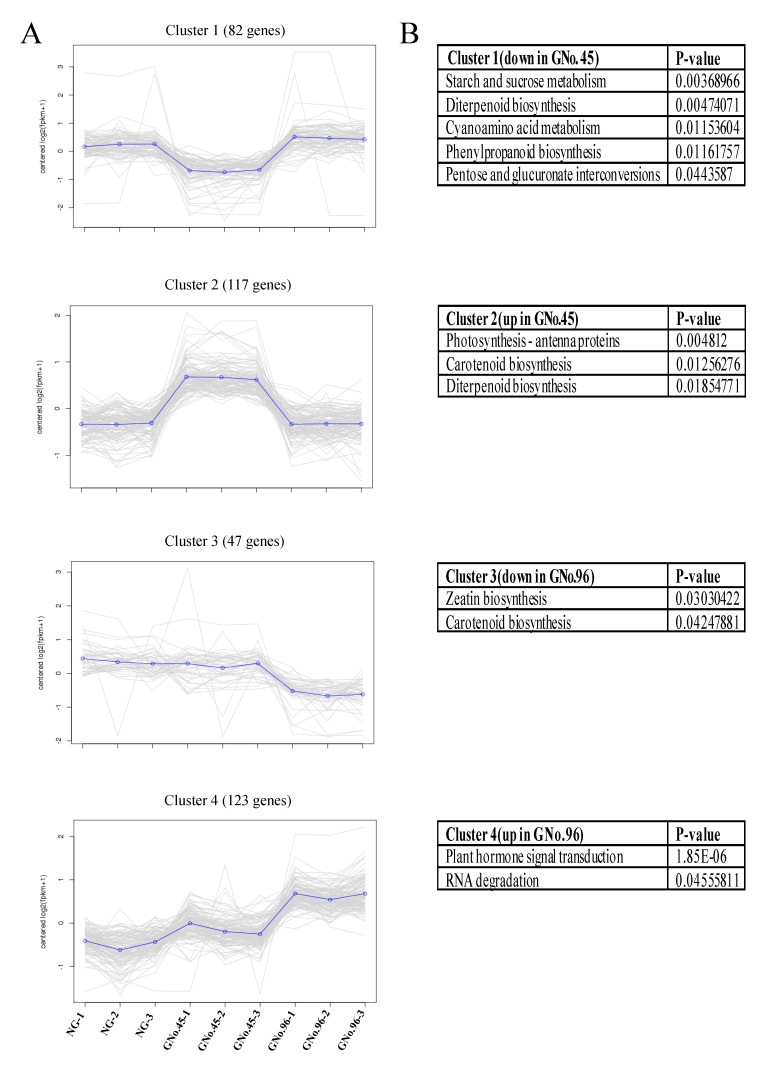
Overview of gene expression clusters calculated by K-means clustering. (**A**) Pearson correlation was used to identify four clusters involving a total of 369 genes with significant expression profile changes in at least one sample. (**B**) GO analysis of DEGs in each cluster, *p* < 0.05 (FDR adjusted).

**Figure 6 ijms-20-03592-f006:**
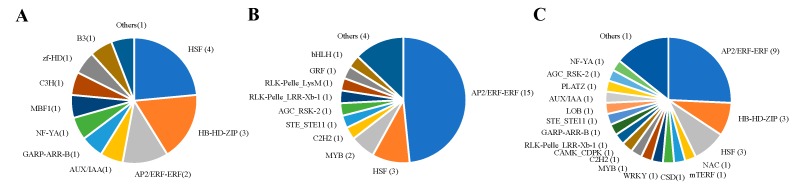
Classification of differently expressed TFs in comparisons of (**A**) GNo.45 vs. NG, (**B**) GNo.96 vs. NG, and (**C**) GNo.45 vs. GNo.96.

**Figure 7 ijms-20-03592-f007:**
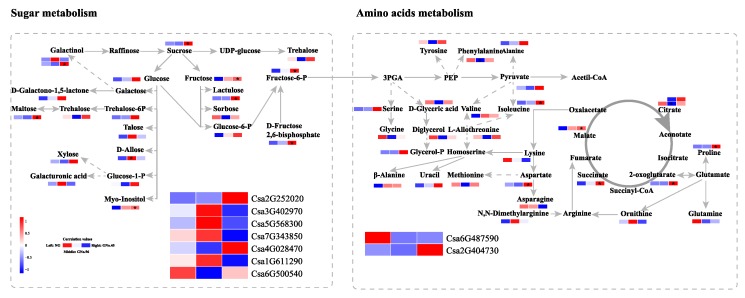
Metabolic profile analyzed by gas chromatography tandem time-of-flight mass spectrometry (GC-TOFMS), and the gene expression pattern analyzed by metabolism-gene correlations. The points within each graph indicate significant differences (*p* < 0.05).

**Figure 8 ijms-20-03592-f008:**
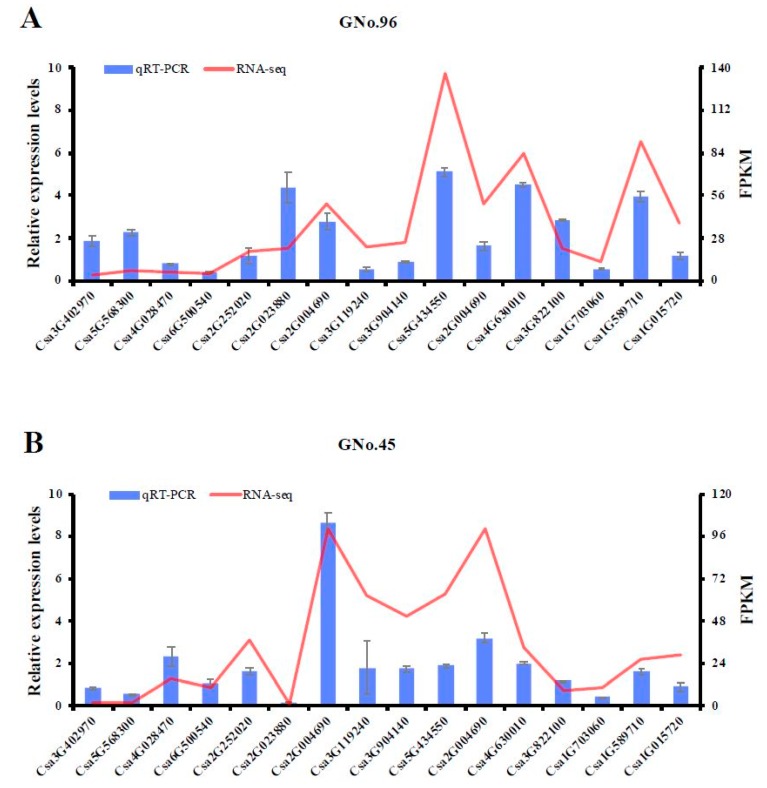
Relative expression levels of 16 genes in cucumber fruits from (**A**) GNo.45 and (**B**) GNo.96, as determined by qRT-PCR and fragments per kilobase of transcript per million (FPKM) mapped reads methods.

**Figure 9 ijms-20-03592-f009:**
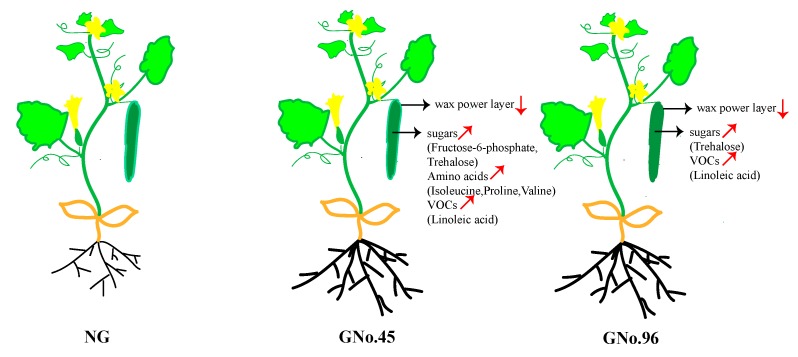
Summary of the differences in metabolites contributing to cucumber flavor in NG, GNo.45, and GNo.96.

**Table 1 ijms-20-03592-t001:** Representative candidate genes determined by different metabolites (DMs)-gene Pearson correlation analysis, which are likely involved in grafted cucumber fruit flavor. NG—non-grafted plants.

Metabolism Classification	Metabolism Name	Candidate Gene Identifier	Log2FC	Functional Annotation	PCC	Combination
Sugars	Fructose-6-phosphate	Csa2G252020	1.04	Glycolysis/gluconeogenesis	0.94045252	GNo.45 vs. NG
		Csa2G252020	1.04	Biosynthesis of amino acids	0.94045252	GNo.45 vs. NG
		Csa2G252020	1.04	Carbon metabolism	0.94045252	GNo.45 vs. NG
		Csa2G252020	1.04	Pentose phosphate pathway	0.94045252	GNo.45 vs. NG
		Csa2G252020	1.04	Fructose and mannose metabolism	0.94045252	GNo.45 vs. NG
		Csa2G252020	1.04	Carbon fixation in photosynthetic organisms	0.94045252	GNo.45 vs. NG
	Trehalose	Csa3G402970	1.09	Starch and sucrose metabolism	−0.9259965	GNo.45 vs. GNo.96
		Csa5G568300	2.05	Starch and sucrose metabolism	−0.9785609	GNo.45 vs. GNo.96
		Csa7G343850	1.42	Starch and sucrose metabolism	−0.8501167	GNo.45 vs. GNo.96
		Csa4G028470	−1.54	Starch and sucrose metabolism	0.99523926	GNo.45 vs. GNo.96
Fatty acids		Csa1G611290	1.12	Starch and sucrose metabolism	−0.9745222	GNo.45 vs. GNo.96
		Csa6G500540	−1.04	Starch and sucrose metabolism	0.98015624	GNo.45 vs. GNo.96
Amino acids	Linoleic acid	Csa2G023880	1.53	Linoleic acid metabolism	−0.9748707	GNo.45 vs. NG
			4.32	Linoleic acid metabolism	0.99350244	GNo.96 vs. NG
	Isoleucine	Csa6G487590	−1.04	Biosynthesis of amino acids	−0.864884	GNo.45 vs. NG
		Csa2G404730	1.11	Biosynthesis of amino acids	0.96014285	GNo.45 vs. NG
		Csa2G252020	1.04	Biosynthesis of amino acids	0.94045252	GNo.45 vs. NG
	Proline	Csa6G487590	−1.04	Biosynthesis of amino acids	−0.864884	GNo.45 vs. NG
		Csa2G404730	1.11	Biosynthesis of amino acids	0.96014285	GNo.45 vs. NG
		Csa2G252020	1.04	Biosynthesis of amino acids	0.94045252	GNo.45 vs. NG
	Valine	Csa6G487590	−1.04	Biosynthesis of amino acids	−0.864884	GNo.45 vs. NG
		Csa2G404730	1.11	Biosynthesis of amino acids	0.96014285	GNo.45 vs. NG
		Csa2G252020	1.04	Biosynthesis of amino acids	0.94045252	GNo.45 vs. NG

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
