# Peer review of "Integrated Metabolome and Transcriptome Analysis Provide Insights into the Effects of Grafting on Fruit Flavor of Cucumber with Different Rootstocks"

_ijms, 2019, doi:10.3390/ijms20143592_

Reviewer 1 Report

Dear authors,

thank you very much for providing us with a revised version addressing all the comments suggested. The manuscript has been significantly improved and reached the level for publication in IJMS. I wish you all the success with your research!

Author Response

Dear professor,

Thanks for your kind suggestions.

I learn a lot through this revision. Thanks for your blessing.

Reviewer 2 Report

The Ms appears interesting but it is written rather badly, with gross spelling errors in English (eg line 132), it lacks a clear and easy conclusion, and a comment on the quality of cucumbers produced by grafted plants and which of the DMs is (are) more relevant.

Moreover:

M&M are really poor; authors should indicate at least the plant age at grafting, and that cucumbers are harvested at a commercial maturity not at a ripening stage;

line 116: Cucurbita moschata must be in Italics;

Figure 2: Authors must indicate clearly the samples’ names and keep them throughout the MS, e.g “NG” for non-grafted, “GNo.45” for grafted on No.45 plants, “GNo.45” for grafted on No.45 plants;

Figure 2 A and B: dimensional bars are necessary;

line 147: Figure 8 is not correct, it is Figure 3;

Figure 3: it is really confused; a) please separate Fig. A, B, e C from Fig. D; b) for D the heat map must include the three samples / three heat map columns for all metabolites;

Figure 4: if red and blue colors indicate up regulation or down regulation, the two colors cannot be employed to indicate the three different samples;

Figure 7: Authors must indicate the three samples above the three columns of the heat maps; finally, the figure is too small.

Author Response

Dear professor,

Thanks for your kind suggestions.

How does grafting affect cucumber fruit flavor? it is an interesting scientific question and deserves further study. Limited by technical means, the study of cucumber fruit flavor affected by grafting was weak, and most research focus on the rootstocks from different genera, such as Cucurbita ficifolia vsCucurbitamoschata,Cucurbita ficifolia vsCucurbita pepo.From the perspective of rootstock breeding, we select the two different rootstocks from the same genera (Cucurbitamoschata), and applied an integrated metabolomics and transcriptomics tools to explore candidate genes involved in grafted cucumber flavor production through metabolomic-gene correlation analysis. Finally, Ten DEGs involved in sugar metabolism (fructose-6-phosphate and trehalose), linoleic acid, and amino-acid (isoleucine, proline, and valine) biosynthesis were identified as candidate genes by DMs-gene correlation analysis, which possibly play important roles in grafted cucumber fruit. The key genes or most relevant DMs should be further studied, and it is carried out now. This study provides valuable information for fruit flavor changes in grafted cucumber by different rootstocks from the same species.

I am sorry to causing any confused and frustrated for you because of my poor English, so the manuscript was re-edited by a native English speaker again. 

We have revised all the comments suggested by you, and I learn a lot through this revision. Please see the attachment.

Thanks for your kind suggestions again.

Round  2

Reviewer 2 Report

The only pont of the present version is the Figure 7 with two boxes too small. I suggest putting the sugar metabolism frame over the Amino acid frame to have a more large column. 

This manuscript is a resubmission of an earlier submission. The following is a list of the peer review reports and author responses from that submission.

Round  1

Reviewer 1 Report

While considering the publication of this report in IJMS   I note what the Authors wrote about the purpose of this study:

'However, the precise mechanism of the effect of grafting on the fruit flavor of cucumber is still not clear, especially concerning that of the rootstocks from the same species.'

Now, the question is, has any of this been achieved? It seems as if the Authors tried to impress us with a wealth of data and sophisticated technologies. But, reading the article left me totally confused and frustrated.

To begin with; the article should have started with a concise list of the major constituents of cucumber flavor, which are probably well known. This would lead to a much more focused research.

Then, has any key flavor metabolite/constituent differing between the grafting treatments been identified?

Furthermore, did the presumed metabolites originate in the rootstocks, or has the grafting induced changes in the metabolism of the scion?

The article does not provide clear answers to any of those questions.

As for now I do not see the point in publication of this overcrowded article in IJMS.

Author Response

While considering the publication of this report in IJMS   I note what the Authors wrote about the purpose of this study:

'However, the precise mechanism of the effect of grafting on the fruit flavor of cucumber is still not clear, especially concerning that of the rootstocks from the same species.'

Answer:This is our scientific question.

Now, the question is, has any of this been achieved? It seems as if the Authors tried to impress us with a wealth of data and sophisticated technologies. But, reading the article left me totally confused and frustrated.

Answer:We partly answer about our scientific question, that is due to the precise mechanism of the effect of grafting on the fruit flavor of cucumber is a complicated question, we can’t achieve this just through the transcriptome and metabolic analysis. We found that grafting significantly changed
the content of sugar and VOCs precursor linoleic acid in fruits, especially in XT/No.45, and also changed the content of amino acids. The related specific genes were identified. To understand this article better, we added the Figure 1. I don’t agree with you that we used wealth data and sophisticated technologies to impress the readers. Because the technical means adopted in this experiment were common and not complicated, and it is also necessary for our research purpose. If you have not fully understood the logic of this article, I am very sorry to make you read my article confused because of my poor English, I have edited it by through academic company. In order to facilitate you to better understand the overall idea of this paper, please see the Figure 2.

To begin with; the article should have started with a concise list of the major constituents of cucumber flavor, which are probably well known. This would lead to a much more focused research.

Answer: This a good suggestion, I have regulated the contents as your suggestion. I added the Figure 3 in the article which briefly summary the major constituents of cucumber flavor in the introduction parts.

Then, has any key flavor metabolite/constituent differing between the grafting treatments been identified?

Answer: Yes, we have found fructose-6-phosphate, trehalose, linoleic acid, isoleucine, proline, and valine were the main flavor metabolite/constituent differing between the grafting treatments. But we can’t confirm which was the most necessary metabolite in the grafted cucumber fruit. That’s
because the evaluation and definition of vegetable flavor is not clear so far. Our article will provide reference information which possibly be the key flavor metabolite/constitute differing between the grafting treatments, but that needs to be verified.

Furthermore, did the presumed metabolites originate in the rootstocks, or has the grafting induced changes in the metabolism of the scion?

Answer: This is a very good question, we measured the metabolites in the grafted cucumber fruit, because metabolites are not specific, we can’t judge the metabolites origin. But we analyze the pumpkins genes in the grafted cucumber fruit, because the cucumber and pumpkin genes were highly homologous, we get some pumpkin genes in cucumber fruit, we are not sure its origins, so we lost these results, we do this work as the followed methods. The specific mechanism needs more study, we can’t get clear answer just through the transcriptome and metabolic analysis.

The process of identified pumpkin genes in the cucumber fruit: In order to find whether there is any pumpkin gene that expressed in scion, we compared the scion expressed genes with a pumpkin unique gene set, which excludes ambiguous gene that can be both mapped to cucumber genome and pumpkin genome.

To generate the pumpkin unique gene set, we download 38 RNA-seq data of various cucumber tissues generated from previous studies. Then the raw reads were mapped to pumpkin (Cucurbita moschata) genome using hisat2 and the normalized expression level was calculated using Stringtie.
Gene with 0 FPKM and 0 TPM in all the 38 RNA-seq data is classified as pumpkin unique gene, while gene with any level of expression is considered an ambiguous gene. Finally, the pumpkin unique genes were selected with python script and made up for the pumpkin unique gene set. Totally,
there are 22,162 genes in the pumpkin unique gene set, consisting more than 2/3 of all pumpkin genes (32,205).
Likewise, we calculated the gene expression level of scion using the same pumpkin reference and analysis pipeline. We found more than 800 pumpkin gene has detectable expression level in scion (in both control and experiment group, see table 1), however, only four of these genes belong to the
pumpkin unique gene set. The four genes were found in XT/XT-1, XT/XT-2 and XT/No.96-3 respectively and were not detected in the replicas of the same sample (see table 2). Therefore, it’s difficult differentiated the expressed pumpkin gene in scion samples was a real pumpkin gene that transmitted upward after grafting or actually a cucumber gene that accidentally mapped to pumpkin genome due to its high similarity to the pumpkin sequences.

The article does not provide clear answers to any of those questions.
Revise: We think the questions you suggested are valuable. We just provided the candidate genes or metabolics, which play important roles in the flavor of cucumber. We can’t get the clear answers just through the analysis of transcriptome and metabolic, it needs the more research. As for now I do not see the point in publication of this overcrowded article in IJMS.

 Reviewer 2 Report

Dear authors,

the manuscript is very well written with an excellent and elucive use of the English language.

The abstract is very well written addressing all the major components namely

1.       the background of the study 

2.       a clear hypothesis 

3.       the key components of the methodology 

4.       the key results  and

5.       a short description of the interpretation/conclusion in the last sentence 

The introduction part is very well written adreesing all the aspects of the hypothesis tested

The MM and methods section provides all the necessary information required for the experimental conditions as well as all the analytical methods used.

The resultsn section clearly described all the data and the statistical analysis provided

The data are verty well interpreted and the reference list covers the relevant literature adequately and in an unbiased way.

Some minor comments/corrections can be found in the attached PDF

Author Response

Revised: Line 40:Cucumber------cucumber
Line 66: regulates ------ regulating
Line 95-97: Two varieties from 102 Cucurbita moschata varieties of rootstock basing on the grafted cucumber growth characteristics, physiology traits, and fruit quality, were selected and the grafted cucumber fruit flavor was largely different------We selected NO.45 and No.96 from 102 Cucurbita moschata varieties of rootstock depending on the grafted cucumber growth characteristics, physiology traits, and fruit quality, and the fruit flavor of cucumber grafted onto the two rootstocks was largely different.
Line 97: Basing on these results------According to these results
Line 99: Basing on metabolomic-gene correlation analysis------through metabolomicgene correlation analysis
Line 108-110: Our findings provide further insights into the impact of different rootstocks on the genetic and physiological characteristics of cucumber flesh and will facilitate the future research efforts to improve the fruit flavor of grafted cucumber plants.------- Our findings provide further insights into the impact of different rootstocks on the genetic and physiological characteristics of cucumber flesh and will facilitate the future research efforts to improve the fruit
flavor of grafted cucumber plants.
Line 206-208: This result indicated that carbon metabolism and plant hormone signals might play crucial roles in cucumber fruit formation which grafted onto different rootstock compared with non-grafted cucumber.-------- This result indicated that carbon metabolism and plant hormone signals might play crucial roles in cucumber fruit formation which grafted onto different rootstock compared with non-grafted cucumber.
Line 291: basing----depending
Line 318: Basing-----According to
Line 332: We presumably the VOCs composition were changed possiblely in cucumber grafted onto Cucurbita moschata No.45 and No.96 rootstocks. ----- The VOCs compositions were changed possiblely in cucumber grafted onto Cucurbita moschata No.45 and No.96 rootstocks.